# Which character strengths may build organizational well-being? Insights from an international sample of workers

Nicole Casali[1]*, Tommaso Feraco[2]

1 Department of Criminology, Max Planck Institute for the Study of Crime, Security and Law, Freiburg, Germany, 2 Department of General Psychology, University of Padova, Padova, Italy

* n.casali@csl.mpg.de

**Data Availability Statement:** There are legal restrictions on sharing a de-identified data set, namely that the data are owned by a third-party organization (i.e., the VIA Institute on Character)

## Abstract

The nature and composition of well-being has been the subject of ongoing debate in the field of positive psychology. Recent discussions identify Seligman's PERMA dimensions as concrete pathways to achieve subjective well-being, rather than a distinct type of well-being. Four additional "building blocks" have been categorized to define positive functioning at work (PERMA+4). The present study adds another level of inquiry, by newly examining the relationships of character strengths and a general factor or character with PERMA+4 and life satisfaction in a large international sample of 5,487 employees. We found that 21 of the 24 character strengths were significantly (yet only slightly) more strongly correlated with PERMA+4 than with life satisfaction, and that PERMA+4 was consistently related to life satisfaction. The happiness strengths (hope, gratitude, zest, curiosity and love) and the general factor of character were also directly and indirectly related to life satisfaction. Taken together, these results suggest that the PERMA+4 dimensions may help bridge the gap between strengths and well-being, explaining how individuals with good character are also those who report higher life satisfaction. Future longitudinal studies should build on the present findings and examine whether character strengths can act as the "building blocks of the building blocks" of life satisfaction.

## Introduction

In the field of positive psychology, there has long been a discussion about well-being: What is it? How can it be defined? What are its components, and the psychological mechanisms that enable it? Several theories have been proposed over the past four decades, with recent models focusing on specific populations, such as employees, to frame well-being in the work domain. In fact, according to Seligman and Csikszentmihalyi' [1], positive psychology is precisely "the science of positive subjective experience [such as subjective well-being], positive individual traits [like the 24 character strengths proposed by Peterson & Seligman [2]], and positive institutions [e.g., organizations]" (p. 1). In this paper, we will attempt to bridge these three levels of inquiry and advance the hypothesis that character strengths are dispositional characteristics that support components of well-being and ultimately lead to life satisfaction.

that has imposed such restrictions. The data requests can be sent to the VIA Institute on Character at research@viacharacter.org. Others would be able to access these data in the same manner as the authors; the authors did not have any special access privileges that others would not have.

**Funding:** The author(s) received no specific funding for this work.

**Competing interests:** The authors have declared that no competing interests exist.

## Theories of well-being

Subjective well-being (SWB), as described by Diener et al. [3], is defined as a hedonic type of well-being that focuses on the individual's evaluation of what's pleasurable. It consists of a cognitive component (satisfaction with life) and two affective components (positive and negative affect). Since its conceptualization, this construct has been at the forefront of positive psychology research, and it has been investigated in thousands of studies. It is commonly considered in the literature as distinct from psychological well-being (PWB), which instead represents the eudaimonic type of well-being, i.e., a focus on growth and optimal psychological functioning, with a greater attention to the interpersonal dimension [4]. For example, Ryff's [5] model of PWB includes six dimensions: self-acceptance, environmental mastery, positive relationships with others, autonomy, purpose in life, and personal growth.

More recently, Seligman [6] proposed the so-called PERMA framework (Positive Emotion, Engagement, Relationships, Meaning, and Accomplishment) as a way to more comprehensively capture the nature of well-being and thereby also reconcile the distinction between psychological and subjective well-being. This model has been saluted as a parsimonious yet exhaustive way to study well-being and has led to a thriving literature applied to diverse contexts and populations [7–12].

## PERMA: The "Building Blocks" of well-being

However, some authors have questioned the need for yet another theory of well-being, as well as its separation from subjective well-being [13]. Using various statistical techniques (including both confirmatory and exploratory factor analysis, as well as cluster analysis), Goodman and colleagues [13] provided strong evidence of an overlap between the two models of well-being, leading them to conclude that PERMA does not represent a distinct, new type of well-being. This study sparked some debate about the nature of PERMA and prompted Seligman himself to respond to these critiques [14]. In his response, the author clarified that PERMA does not in fact constitute a new type of well-being, but rather the "building blocks" (as he calls them) that lead to SWB, which he sees as a "useful final common path of the elements of well-being (that are PERMA dimensions)" (p. 1). In other words, PERMA can be seen as a useful guide for building well-being and developing interventions to achieve the ultimate goal of being satisfied with one's life conditions. Seligman also proposed some criteria to expand and evaluate prospective elements of well-being (e.g., contribute to well-being, be pursued for their own sake). As a result, Donaldson et al. [15, 16] have proposed the Positive Functioning at Work model, also known as PERMA+4, as it adds four additional dimensions to Seligman's PERMA framework (physical health, mindset, environment, and economic security). This model aims to strengthen the PERMA framework by adapting it to the work domain, thus providing a more holistic picture of organizational well-being (see S1 Table for an overview of the above models). In their study of knowledgeable co-worker pairs, Donaldson and colleagues [15] found that self-reported PERMA significantly predicted both self- and other-reported SWB; the same was true for self-reported PERMA+4, and both other-reported PERMA and PERMA+4. These findings support the idea that PERMA (+4) dimensions may indeed be better understood as foundational elements of well-being and clear pathways towards it, rather than new types of well-being to be studied separately.

## Character strengths: The building blocks of the building blocks?

Character strengths are 24 positive, trait-like individual qualities that were theorized by Peterson and Seligman [2] as the psychological ingredients that constitute and lead to six higher-order, more abstract moral virtues (wisdom and knowledge, courage, humanity, justice, temperance, and transcendence, see S2 Table for an overview); although morally valued on their

 

own right, strengths are bound to produce positive outcomes, and account for the good life. Recently, some authors proposed an additional level of analysis, namely a general factor of character (see the next paragraph) that describes the general dispositional positivity of individuals [17–19]. Literature studies have confirmed the important role of single strengths for well-being time and again [see 20, 21 for an overview] with evidence for the relevant effect of a general factor of character [17, 22]. In addition, recent theorizations [23] have detailed various functions that character strengths play in helping individuals thrive. Specifically, character strengths have a strong positivity effect, helping us to take advantage of and optimize opportunities: priming them (preparing us to use our best qualities when the situation calls for them); being present to them through mindfulness (thus helping us to balance our resources, and adapt them to the situation); and appreciating them (acknowledging their value after they occurred). Consistent with Fredrickson's [24] broaden-and-build model, character strengths expand our skill set and support our positive response to what's present, while also helping us build resources for future opportunities. Put another way, these 24 strengths may represent the foundations of well-being because they are trait-like, relatively stable [25], universal qualities that can be cultivated to increase well-being [26]. But how is this possible? Which are the specific mechanisms connecting strengths with well-being? This is still an open but fundamental question [26, 27]. We speculate that the PERMA+4 dimensions (the building blocks of work-related well-being in Seligman's view) may represent such pathways and may help us understand how building strengths can also lead to greater life satisfaction. The workplace represents one of the levels of inquiry originally identified by Seligman and Csikszentmihalyi'[1] in their manifesto on positive psychology. The workplace is a targeted yet highly relevant domain of life that has been shown to offer multiple opportunities for individuals to realize their potential and achieve a sense of purpose and meaning [28–31]. Organizational well-being therefore here refers to both the eudaimonic (PERMA+4 dimensions) and hedonic (life satisfaction) components of well-being experienced by workers, in line with established frameworks [32]. These different kinds of well-being represent "what" constitutes organizational well-being, while strengths have to do with "how" to enhance well-being in the workplace [32]. In this sense, character strengths could be considered as the "building blocks of the building blocks" of well-being, or, put differently, as more distal predictors of life satisfaction compared to PERMA+4. Looking more closely at the criteria proposed by Seligman [14] for assessing potential elements of well-being, we could argue that character strengths meet most of them. First of all, there is accumulating evidence that character strengths are significantly associated with both life satisfaction [13, 33] and PERMA [12, 13], which would fulfill the criterion "contribute to well-being". Then, for the criterion "to be pursued for their own sake (rather than as means to an end)", strengths are, by definition, intrinsically morally valued and do not depend on potential positive outcomes to be classified as such [2], as also shown empirically [34]. Although the classification has been subject to criticism and revision [35], it was also developed after intensive historical, cross-cultural, and philosophical review, making it rather comprehensive, which would speak in favor of the taxonomy being "exclusive and exhaustive". As briefly mentioned, there are a variety of character strengths based interventions [27, 35], and there is strong evidence for their efficacy on life satisfaction [26], with some encouraging evidence of their ability to increase strengths trait levels [36]. This evidence would support that character strengths can indeed be translated into specific interventions to build each other as well as well-being. Although 24 traits are quite a large number of variables to be considered, strengths can be considered as a unique factor [17, 22], as recently reviewed by McGrath (18, and see next paragraph for a more detailed description), making the list more parsimonious. Lastly, although strengths are highly correlated with each other, they can be assessed separately using the VIA-IS questionnaires [2] and have shown differential correlations with well-being indicators. As such, they can be defined and measured independently.

## A general factor of character?

Similar to other fields (e.g., intelligence, personality, or psychopathology), there has been increasing discussion of the possibility of a general factor of character, i.e., a common core shared by the specific character strengths. These two levels of analysis have implications that are theoretical (they tap into different hierarchical layers of character), statistical (they can help reduce the number of variables to be considered and disentangle their specific roles), and practical (they can suggest what is most important and thus should be targeted in interventions). It has been shown that a bifactor model, in which items load on both a general factor of character and on the corresponding specific strength, can indeed represent the data well [17, 19], and that the general factor has incremental predictive validity over the specific character strengths factors, at least for life satisfaction [17].

## Rationale and hypotheses

The present study aims to make an initial contribution to the recent debate on well-being and its constitutive elements, or "building blocks" [13, 15, 16] by proposing to consider an additional level of analysis, namely the role of character strengths. To rigorously examine the relationships among these three constructs, we will use a two-stage approach (see the Data Analysis section) and a large international sample of employees.

We will test the following hypotheses:

- Hypothesis 1: Character strengths (in terms of the 24 strengths and the general factor of character) will be positively related to both PERMA+4 total score and life satisfaction. Previous studies only examined the relationship between the single 24 strengths and Seligman's original PERMA dimensions [12, 13] and found that hope, love, gratitude, and zest were the strengths with the highest correlations; similarly, Bruna et al. [33] meta-analysis evidenced zest, hope, gratitude, curiosity, and love as the strengths most strongly related to life satisfaction. Goodman et al. [13] reported very similar (but slightly stronger for PERMA) correlations of strengths with PERMA and SWLS, once again finding gratitude, hope, and zest as the strengths most strongly related to well-being. We aim to extend these findings to the PERMA+4 dimensions and to a different level of analysis (i.e., the general factor of character), which has recently proposed as a valid measure of character [17, 22]. These relationships would provide a first indication of the role of strengths as building blocks (of the building blocks) of life satisfaction;

- Hypothesis 2: PERMA+4 will be positively related to SWB, as previously reported by Donaldson and colleagues [15, 16], further supporting the notion that it may represent the building block of life satisfaction;

- Hypothesis 3: PERMA+4 will mediate the relationship between strengths and SWLS, i.e., when controlling for the effect of PERMA+4 dimensions, the relationship between character strengths and life satisfaction may disappear.

Collectively, our hypothesis would provide a first indication in favor of these relationships, although they would not reveal any causality.

## Method

### Participants and procedure

Data from 14364 participants were kindly collected by the VIA Institute on Character, by asking individuals filling out the VIA-IS-P through the Institute's website to also complete the

Satisfaction With Life Scale (SWLS) and the PERMA+4 (in this order), and then to provide standard demographic information. No IRB approval was required according to the University of Padova Ethics Committee, since the data was collected through an independent, international body (i.e., the VIA Institute). Although no informed consent was formally required, participants were informed of the VIA Institute's privacy policy (https://www.viacharacter.org/privacy-policy, which they could access at any time by clicking at the bottom of each page) and could opt out of the survey at any moment and skip any questions they did not feel comfortable answering. For instance, they were informed that if they opted in to participate in the study, the VIA Institute could share the de-identified results (i.e., without the participant's name or email address) with the researchers. To the aim of the present study, data were retained only if participants completed both SWLS and PERMA+4, as well as if they indicated to be employed, as PERMA+4 specifically addresses the work environment. Our final sample thus consists of 5487 employees. The majority was employed full-time (80%), while the remaining (20%) was employed part-time (i.e., up to 39 hours per week). Participants worked in a variety of sectors: 1323 (24.1%) in business and administration, 817 (14.9%) in education and teaching, 698 (12.7%) in STEM professions, 564 (10.3%) in health professions, 270 (4.9%) in the military, 194 (3.5%) in counselling, 83 (1.5%) in legal professions, and the remaining 1455 (26.5%) in other, not specified areas; 293 (5.3%) did not report their occupation sector. Among the participants, 944 (17.2%) were in the age range between 18 and 24, 1515 (27.6%) between 25 and 34, 1411 (25.7%) between 35 and 44, 1095 (20%) between 45 and 54, 451 (8.2%) between 55 and 64, 66 (1.2%) between 65 and 74, and 5 (0.1%) older than 75. As for education, 88 (1.6%) did not complete high school, 1142 (20.8%) had a high school diploma, 1998 (36.4%) had a Bachelor's degree, 1269 (23.1%) had a Master's degree, 235 (4.3%) had a Ph.D. degree or above, and 721 (13.1%) an associate/professional degree; 34 (0.6%) did not report their education. Gender was not available.

## Materials

The VIA-IS-P [37] measures the 24 character strengths. This consists of 96 positively keyed items (four for each strength) scored on a 5-point Likert scale (1 = very much unlike me; 5 = very much like me). Examples of items include "I always speak up in protest when I hear someone say mean things" for bravery, or "I am an extremely grateful person" for gratitude. The measure displayed satisfactory internal consistency for both strengths (Cronbach's α ranging from .65 for humility to .87 for love) and virtues (Cronbach's α ranging from .62 for temperance to .79 for transcendence). Similarly, satisfactory properties were found in the present sample for strengths (Cronbach's α ranging from .67 for judgement to .87 for love), virtues (Cronbach's α ranging from .81 for justice to .87 for transcendence) and character overall score (α = .95).

The Positive Functioning at Work Scale (16) measures the PERMA+4 dimensions. This consists of 29 items, evaluating nine dimensions on a 7-point Likert scale (1 = strongly disagree, 7 = strongly agree): *Positive emotions* (three items, e.g., "I feel joy in a typical workday"); *engagement* (three items, e.g., "I lose track of time while doing something I enjoy at work"); *relationships* (four items, e.g., "I can receive support from coworkers if I need it"); *meaning* (three items, e.g., "My work is meaningful"); *accomplishment* (three items, e.g., "I typically accomplish what I set out to do in my job"); *physical health* (four items, e.g., "I typically feel physically healthy"); *mindset* (three items, e.g., "I believe I can improve my job skills through hard work"); *environment* (three items, e.g., "There is plenty of natural light in my workplace"); and *economic security* (three items, e.g., "I am comfortable with my current income"). The internal consistency of the overall scale is reportedly excellent (α = .94), while the

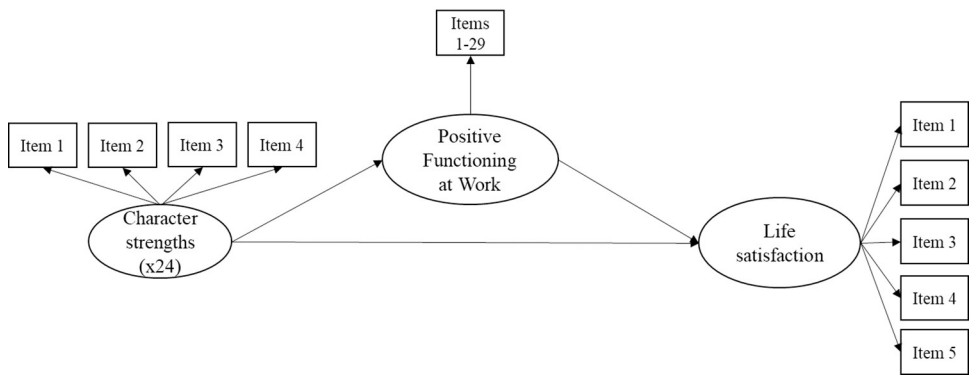

**Fig 1. Graphical representation of the first hypothesized path model (M1).**

reliability for the subscales ranges from .76 for environment to .93 for positive emotions. These results were replicated in our sample for both the overall score ($\alpha$ = .91) and the subscales (alpha ranging from .66 for environment to .93 for positive emotions).

The Satisfaction With Life Scale [3] measures the cognitive component of subjective well-being. This comprises five items scored on a 7-point Likert scale (1 = completely disagree, 7 = completely agree) measuring overall life satisfaction (e.g., "The conditions of my life are excellent"). The scale showed good internal consistency in the original form ($\alpha$ = .87) as well as in the present sample ($\alpha$ = .88).

## Data analysis

To examine the relationships of character strengths and overall character with SWLS and PERMA+4 we first computed the correlation sizes and compared them using Fisher's *z* transformation to get an initial indication of any stronger relationships with PERMA+4 as compared to SWLS. Descriptively, we also computed the correlations between character strengths and each of the PERMA+4 dimensions, interpreting correlations below .20 as modest, between .20 and .30 as small, and above .30 as moderate. We then fitted two path models using the R package lavaan [38], considering the items as ordinal. Figs 1 and 2 show graphical representations of the two models. Model 1 had the 24 character strengths (modelled as latent variables) as predictors, while Model 2 had the general character factor and the specific strengths (i.e., calculated after accounting for the variance explained by the general character factor) as orthogonal predictors. In both models, life satisfaction (modelled as a latent variable) was the dependent variable, and Positive Functioning at Work (also modelled as a unidimensional

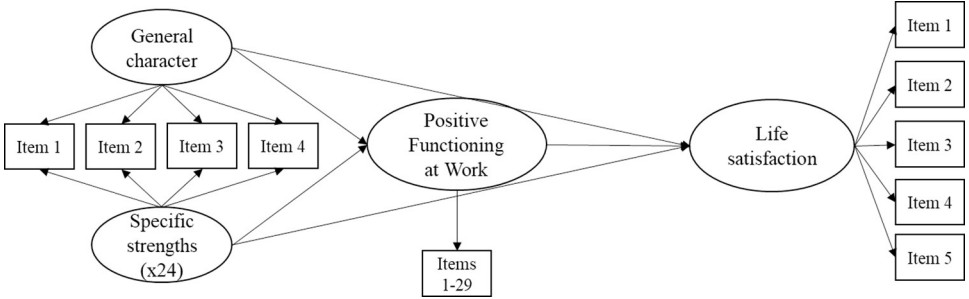

**Fig 2. Graphical representation of the second hypothesized path model (M2).**

**Table 1. Correlations of character strengths and character with life satisfaction and PERMA+4.**

| | α | SWLS | PERMA+4 | \|Δr\| | Test statistic | p |
|---|---|---|---|---|---|---|
| Appreciation of beauty | .80 | .10* | .14* | .04 | 2.13 | .03 |
| Bravery | .73 | .11* | .21* | .10 | 5.38 | < .001 |
| Creativity | .78 | .08* | .17* | .09 | 4.79 | < .001 |
| Curiosity | .70 | .23* | .27* | .04 | 2.23 | .03 |
| Fairness | .82 | .16* | .22* | .06 | 3.26 | < .001 |
| Forgiveness | .71 | .17* | .23* | .06 | 3.27 | < .001 |
| Gratitude | .76 | 48* | .43* | .05 | −3.30 | < .001 |
| Honesty | .70 | .25* | .30* | .05 | 2.83 | < .001 |
| Hope | .73 | .47* | .51* | .04 | 2.76 | .01 |
| Humility | .68 | .07* | .12* | .05 | 2.64 | .01 |
| Humor | .85 | .13* | .18* | .05 | 2.68 | .01 |
| Judgement | .67 | .12* | .19* | .07 | 3.76 | < .001 |
| Kindness | .72 | .11* | .20* | .09 | 4.83 | < .001 |
| Leadership | .83 | .23* | .32* | .09 | 5.10 | < .001 |
| Love | .87 | .26* | .24* | .02 | −1.12 | .26 |
| Love of learning | .80 | .14* | .27* | .13 | 7.12 | < .001 |
| Perseverance | 82 | .25* | .36* | .11 | 6.36 | < .001 |
| Perspective | .79 | .19* | .25* | .06 | 3.30 | < .001 |
| Prudence | .81 | .16* | .19* | .03 | −1.62 | .11 |
| Self-regulation | .79 | .27* | .35* | .08 | 4.64 | < .001 |
| Social intelligence | .71 | .17* | .23* | .06 | 3.27 | < .001 |
| Spirituality | .81 | .20* | .27* | .07 | 3.88 | < .001 |
| Teamwork | .68 | .16* | .28* | .12 | 6.61 | < .001 |
| Zest | .81 | .40* | .48* | .08 | 5.20 | < .001 |
| Character | .95 | .39* | .50* | .11 | 7.20 | < .001 |

*Note.* * p < .001

latent variable) was the mediator. Due to the large sample size obtained in the present study, we only considered standardized betas with associated $p \leq .001$ as significant.

## Results

### Correlations with positive functioning at work and life satisfaction

Table 1 shows the correlations of character strengths and overall character scores with both SWLS and PERMA+4 total scores. Correlations were generally significantly stronger for PERMA+4 than for SWLS, with the exception of gratitude, that correlated more strongly with SWLS. Specifically, correlations with PERMA+4 were significantly stronger for 21 out of 24 strengths, and not significantly different for two strengths, namely love ($p = .26$) and prudence ($p = .11$). The difference in correlation size ranged between .02 for love to .13 for love of learning, with a mean difference of .07 (median difference = .06).

### Correlations with positive functioning at work dimensions

Table 2 shows the correlation of character strengths and overall character with PERMA+4. For positive emotions, three character strengths showed correlations above .30 (zest: $r = .48$, hope: $r = .45$; gratitude: $r = .39$). For engagement, the correlations were mostly very modest, with only love of learning ($r = .29$) and curiosity ($r = .23$) showing correlations above .20. For

**Table 2. Correlations of character strengths, virtues, and character with PERMA+4 dimensions.**

|  | P | E | R | M | A | PH | MI | EN | ES |
|---|---|---|---|---|---|---|---|---|---|
| Appreciation of beauty | .13* | .15* | .09* | .13* | .11* | .06* | .07* | .12* | −.03 |
| Bravery | .20* | .13* | .08* | .18* | .27* | .12* | .15* | .12* | .02 |
| Creativity | .16* | .19* | .04** | .14* | .22* | .08* | .13* | .10* | 0 |
| Curiosity | .23* | .23* | .13* | .18* | .24* | .19* | .20* | .13* | .06* |
| Fairness | .17* | .11* | .17* | .18* | .19* | .12* | .17* | .10* | .06* |
| Forgiveness | .22* | .08* | .20* | .17* | .15* | .16* | .17* | .13* | .06* |
| Gratitude | .39* | .11* | .28* | .34* | .35* | .30* | .31* | .25* | .14* |
| Honesty | .20* | .13* | .18* | .20* | .33* | .22* | .19* | .17* | .13* |
| Hope | .45* | .16* | .32* | .36* | .47* | .38* | .39* | .27* | .17* |
| Humility | .07* | .04** | .07* | .07* | .09* | .12* | .09* | .08* | .04 |
| Humor | .16* | .07* | .14* | .11* | .17* | .14* | .14* | .10* | 0 |
| Judgment | .09* | .12* | .09* | .09* | .23* | .15* | .15* | .07* | .12* |
| Kindness | .19* | .14* | .19* | .19* | .21* | .06* | .17* | .13* | −.04 |
| Leadership | .26* | .15* | .20* | .23* | .36* | .16* | .26* | .17* | .11* |
| Love | .22* | .05* | .21* | .20* | .21* | .13* | .16* | .17* | .02 |
| Love of learning | .22* | .29* | .12* | .21* | .27* | .11* | .20* | .14* | .07* |
| Perseverance | .26* | .07* | .16* | .21* | .45* | .29* | .24* | .20* | .19* |
| Perspective | .16* | .15* | .15* | .19* | .30* | .14* | .18* | .11* | .10* |
| Prudence | .08* | .03* | .12* | .10* | .24* | .16* | .12* | .08* | .16* |
| Self-regulation | .22* | .04** | .16* | .18* | .40* | .35* | .22* | .18* | .21* |
| Social intelligence | .21* | .10* | .19* | .19* | .23* | .13* | .20* | .15* | −.01 |
| Spirituality | .28* | .08* | .16* | .31* | .27* | .11* | .23* | .16* | −.01 |
| Teamwork | .24* | .07* | .33* | .20* | .19* | .14* | .25* | .16* | .05** |
| Zest | .48* | .14* | .27* | .32* | .40* | .38* | .35* | .27* | .15* |
| Character | .42* | .22* | .31* | .37* | .50* | .33* | .37* | .28* | .14* |

*Note.* P = Positive emotions, E = Engagement, R = Relationships, M = Meaning, A = Accomplishment, PH = Physical health, MI = Mindset, EN = Environment, ES = Economic security, * *p* < .001

relationships, two correlations were above .30 (teamwork: *r* = .33, hope: *r* = .32). For meaning, four correlations resulted stronger than .30 (hope: *r* = .36, gratitude: *r* = .34, zest: *r* = .32, and spirituality: *r* = .31). For accomplishment, the correlations were generally stronger, with eight correlations exceeding .30 (with the top three correlated strengths being hope: *r* = .47, perseverance: *r* = .45, and self-regulation: *r* = .40). With respect to the additional four PERMA dimensions, the correlations were mostly small. For physical health, four correlations exceeded .30 (zest: *r* = .38, hope: *r* = .38, self-regulation: *r* = .35, and gratitude: *r* = .30). In terms of mindset, three strengths showed a correlation above .30 (hope: *r* = .39, zest: *r* = .35, and gratitude: *r* = .31). For environment, only four correlations were between .20 and .30 (zest: *r* = .27, hope: *r* = .27, gratitude: *r* = .25, and perseverance: *r* = .20). Finally, the correlations for economic security were very low, except for a small correlation with self-regulation (*r* = .21).

## Character strengths, PERMA+4, and SWLS

Table 3 presents the main results of the first model (M1, see Fig 1). This path model (with the 24 latent character strengths as predictors, life satisfaction as latent dependent variable, and latent positive functioning at work as mediator) showed good fit to the data (CFI = .932, TLI = .928, RMSEA = .044, SRMR = .050). All item loadings were significant, with a mean of .81 for life satisfaction and of .61 for positive functioning at work. Results indicated that only three

**Table 3. Results of the first path model (M1), with character strengths as predictors, positive functioning at work as mediator, and life satisfaction as dependent variable.**

| Predictor | Life satisfaction | | Positive Functioning at Work | | Indirect effect | |
|---|---|---|---|---|---|---|
| | $B$ | CI | $\beta$ | CI | $\beta$ | CI |
| Appreciation of beauty | −.24* | [−.31, −.17] | −.00 | [−.06, .05] | .00 | [−.01, .01] |
| Bravery | −.01 | [−.08, .05] | −.05 | [−.11, .00] | .00 | [−.00, .00] |
| Creativity | −.23* | [−.32, −.14] | −.08 | [−.16, −.01] | .02 | [.00, .04] |
| Curiosity | .63* | [.46, .80] | −.11 | [−.24, .03] | −.07 | [−.16, .03] |
| Fairness | −.02 | [−.10, .06] | .12 | [.05, .19] | −.00 | [−.01, .01] |
| Forgiveness | .01 | [−.08.11] | −.13 | [−.21, −.05] | −.00 | [−.02, .01] |
| Gratitude | 1.13* | [.76, 1.50] | −.45 | [−.76, −.15] | −.51 | [−.98, − .05] |
| Honesty | .04 | [−.02, .09] | .07 | [.02, .12] | .00 | [−.00, .01] |
| Hope | −.52 | [−.96, −.08] | .83* | [.47, 1.20] | −.43 | [−.93, .07] |
| Humility | −.09 | [−.15, −.04] | −.04 | [−.09, .00] | .00 | [.00, .01] |
| Humor | −.02 | [−.04, .07] | −.04 | [−.08, .00] | −.00 | [−.00, .00] |
| Judgment | .00 | [−.18, .19] | −.19 | [−.35, −.03] | −.00 | [−.04, .04] |
| Kindness | −.25 | [−.40, −.11] | .16 | [.04, .27] | −.04 | [−.09, .01] |
| Leadership | .21* | [.15, .28] | .03 | [−.03, .08] | .01 | [−.01, .02] |
| Love | −.01 | [−.06, .05] | .07 | [.03, .11] | −.00 | [−.00, .00] |
| Love of learning | −.24* | [−.33, −.17] | .10 | [.03, .16] | .02 | [−.04, .00] |
| Perseverance | .16 | [.03, .29] | −.08 | [−.19, .03] | −.01 | [−.04, .01] |
| Perspective | −.05 | [−.11, .02] | .08 | [.03, .13] | −.00 | [−.01, .00] |
| Prudence | .08 | [−.10, .25] | .13 | [−.02, .27] | .01 | [−.01, .00] |
| Self-regulation | .04 | [−.03, .12] | −.03 | [−.09, .03] | −.00 | [−.01 .03] |
| Social intelligence | .13 | [−.02, .28] | −.19 | [−.31, −.07] | −.03 | [−.01, .00] |
| Spirituality | −.29* | [−.37, −.21] | .12* | [.05, .18] | −.03 | [−.06, −.01] |
| Teamwork | −.05 | [−.12, .02] | .04 | [−.01, .10] | −.00 | [−.01, .00] |
| Zest | −.28* | [−.39, −.17] | .28* | [.19, .36] | −.08 | [−.12, −.03] |
| Positive Functioning at Work | .48* | [.43, .53] | - | | - | |
| $R^2$ | .66 | | .41 | | - | |

*Note.* SWLS = Satisfaction With Life Scale, PERMA+4 = Positive Functioning at Work Scale. $\beta$ = standardized beta coefficient, CI = 95% Confidence Intervals, $R^2$ = percentage of variance explained, * = $p < .001$

out of 24 character strengths were significantly positively related to life satisfaction, namely gratitude ($\beta$ = 1.13), curiosity ($\beta$ = .63), and leadership ($\beta$ = .21). In turn, positive functioning at work was also significantly positively associated with life satisfaction ($\beta$ = .48). Three strengths, namely hope ($\beta$ = .83), zest ($\beta$ = .28), and spirituality ($\beta$ = .12), were positively associated with positive functioning at work. We also observed small, negative relationships of spirituality ($\beta$ = −.28), zest ($\beta$ = −.28), love of learning ($\beta$ = −.24), appreciation of beauty ($\beta$ = −.24), and creativity ($\beta$ = −.23) with life satisfaction. None of the indirect effects of character strengths on SWLS through the mediation of positive functioning at work were significant. The model explained 66% of the variance in SWLS, and 41% of variance in positive functioning at work. We also fitted a multiple linear regression model with the 24 character strengths and PERMA+4 as independent variables and SWLS as dependent variable and obtained similar results, with hope, love, and zest also showing small positive effects on life satisfaction (see S3 Table). Although the parameters were normally estimated by the model, high collinearity between the strengths may have led to imprecision in the beta estimates, as indicated by the large confidence intervals. This also suggests that it might indeed be useful to model the common variance as a general factor of character, which we did in Model 2.

**Table 4. Results of the second path model (M2), with specific character strengths and general factor of character as predictors, positive functioning at work as mediator, and life satisfaction as dependent variable.**

| Predictor | Life satisfaction | | Positive Functioning at Work | | Indirect effect | |
|---|---|---|---|---|---|---|
| | $\beta$ | CI | $\beta$ | CI | $\beta$ | CI |
| Appreciation of beauty | −.08* | [−.11, −.05] | −.13* | [−.15, −.10] | .01* | [.01, .02] |
| Bravery | −.13* | [−.16, −.10] | −.10* | [−.13, −.08] | .01* | [.01, .02] |
| Creativity | −.16* | [−.19, −.12] | −.16* | [−.18, −.13] | .03* | [.02, .03] |
| Curiosity | −.05 | [−.08, −.02] | −.11* | [−.14, −.09] | .01 | [.00, .01] |
| Fairness | −.04 | [−.06, −.01] | −.05* | [−.08, −.03] | .00 | [.00, .00] |
| Forgiveness | −.03 | [−.05 .00] | −.05* | [−.07, −.02] | .00 | [.00, .00] |
| Gratitude | .44* | [.41, .47] | .10* | [.07, .12] | .04* | [.03, .06] |
| Honesty | .06 | [.03, .09] | .00 | [−.02, .03] | .00 | [−.00, .00] |
| Hope | .37* | [.33, .41] | .17* | [.14, .20] | .06* | [.05, .08] |
| Humility | −.08* | [−.11, −.05] | −.12* | [−.13, −.08] | .01* | [.00, .01] |
| Humor | −.04 | [−.06, .01] | −.05* | [−.07, −.02] | .00 | [.00, .00] |
| Judgment | −.06* | [−.09, −.03] | −.10* | [−.12, −.07] | .01 | [.00, .01] |
| Kindness | −.16* | [−.20, −.12] | −.12* | [−.14, −.09] | .02* | [.01, .03] |
| Leadership | −.00 | [−.03, .03] | −.02 | [−.04, .01] | .00 | [.00, .00] |
| Love | .08* | [.05, .11] | −.03 | [−.05, −.01] | .00 | [−.00, .00] |
| Love of learning | −.09* | [−.12, −.06] | −.03 | [−.05, −.01] | −.00 | [.00, .01] |
| Perseverance | .05 | [.02, .08] | .07* | [.05, .09] | .00 | [.00, .01] |
| Perspective | −.04 | [−.15, −.01] | −.07* | [−.10, −.05] | .00 | [.00, .00] |
| Prudence | .01 | [−.06, .01] | −.06* | [−.08, −.04] | −.00 | [−.00, .00] |
| Self-regulation | .09* | [.05, .11] | .05* | [.02, .07] | .00 | [.00, .01] |
| Social intelligence | −.11* | [−.15, −.07] | −.15* | [−.17, −.12] | .02* | [.01, .02] |
| Spirituality | −.03 | [−.05, .00] | −.04 | [−.06, −.02] | .00 | [.00, .01] |
| Teamwork | −.04 | [−.07, −.02] | .02 | [−.01, .04] | −.00 | [−.00, .00] |
| Zest | .23* | [.20, .26] | .18* | [.15, .20] | .04* | [.03, .05] |
| General factor of character | .34* | [.29, .40] | .57* | [.55, .60] | .20* | [.16, .23] |
| PERMA+4 | .15* | [.08, .23] | - | | - | |
| R² | .80 | | .55 | | - | |

*Note.* SWLS = Satisfaction With Life Scale, PERMA+4 = Positive Functioning at Work Scale. $\beta$ = standardized beta coefficient, CI = 95% Confidence Intervals, R² = percentage of variance explained, * = $p < .001$

Table 4 shows the results of the second model (M2, Fig 2). This path model (with the 24 specific character strengths and the general factor of character as predictors, life satisfaction as dependent variable, and positive functioning at work as mediator) showed a descriptively poorer fit to the data (CFI = .898, TLI = .894, RMSEA = .089, SRMR = .076). All item loadings were significant, with a mean of .81 for life satisfaction, of .61 for positive functioning at work, and of .43 for the general factor of character. The amount of variance in the specific strengths that was explained by the general factor ranged from .31 (hope) to .85 (prudence). Results indicated positive, significant associations with life satisfaction for the general factor of character ($\beta$ = .34) and five of the specific strengths of gratitude ($\beta$ = .44), hope ($\beta$ = .37), zest ($\beta$ = .23), self-regulation ($\beta$ = .09), and love ($\beta$ = .08). We also observed small, negative effects on life satisfaction for the specific strengths of kindness ($\beta$ = −.16), creativity ($\beta$ = −.16), bravery ($\beta$ = −.13), social intelligence ($\beta$ = −.11), love of learning ($\beta$ = −.09), appreciation of beauty ($\beta$ = −.08), humility ($\beta$ = −.08), and judgment ($\beta$ = −.06). Positive functioning at work was also positively associated with life satisfaction, although to a descriptively lesser degree than in M1 ($\beta$ =

.15). In turn, the general factor of character showed a strong positive association with Positive Functioning at Work ($\beta$ = .57), followed by small positive associations of five specific strengths, namely zest ($\beta$ = .18), hope ($\beta$ = .17), gratitude ($\beta$ = .10), perseverance ($\beta$ = .07), and self-regulation ($\beta$ = .05). Small negative associations appeared for 13 specific strengths, namely creativity ($\beta$ = −.16), social intelligence ($\beta$ = −.15), appreciation of beauty ($\beta$ = −.13), kindness ($\beta$ = −.12), humility ($\beta$ = −.12), curiosity ($\beta$ = −.11), bravery ($\beta$ = −.10), judgment ($\beta$ = −.10), perspective ($\beta$ = −.07), prudence ($\beta$ = −.06), fairness ($\beta$ = −.05), forgiveness ($\beta$ = −.05), and humor ($\beta$ = −.05). Ten of the indirect effects of character strengths on SWLS through the mediation of positive functioning at work were significant and positive. These regarded overall character ($\beta$ = .20) and the specific strengths of hope ($\beta$ = .06), gratitude ($\beta$ = .04), zest ($\beta$ = .04), creativity ($\beta$ = .03), kindness ($\beta$ = .02), social intelligence ($\beta$ = .02), appreciation of beauty ($\beta$ = .01), bravery ($\beta$ = .01), and humility ($\beta$ = .01). The model explained 80% of the variance in life satisfaction, and 55% of the variance in positive functioning at work.

## Discussion and conclusions

Building well-being in the workplace begins with identifying the pathways that lead to it, and then developing diverse interventions that take advantage of these mechanisms. The more we are able to identify meaningful and alternative ways to build well-being, the greater the number of people we can benefit. Rather than focusing prescriptively on single, one-size-fits-all features, researchers should aim to illustrate descriptively the multiple ways in which individuals can develop their own well-being. In this sense, studying character and PERMA as building blocks of well-being means attempting to advocate for a diversity of possible successful pathways to reach life satisfaction. In this study, we hypothesized that strengths and character would be positively related to the nine elements of well-being identified by Donaldson et al. [15, 16] in their model of positive functioning at work, and that these in turn would positively affect life satisfaction. We respond here to the authors' call to "position PERMA+4 as a process factor, and not an active or targeted antecedent of well-being" (Donaldson et al., [39], p. 9) by placing the focus on character strengths, here considered as potential factors "needed to activate PERMA+4 as a means to enhance work-related well-being" (ibidem).

The results of the correlational analyses provided preliminary evidence in support of this line of reasoning, by showing (slightly) stronger correlations between character strengths and PERMA+4 compared to life satisfaction. These findings are consistent with Goodman et al. [13], who also found small differences in correlation magnitudes between SWLS and the original PERMA measure, favoring the latter. In addition, we examined correlations with the nine dimensions of PERMA+4 separately. Our results showed that strengths were meaningfully related to all nine dimensions, with more and stronger correlations for accomplishment and positive emotions, and fewer and weaker correlations for environment and economic security. Some strengths correlated moderately with several of the PERMA+4 dimensions, while others were descriptively more related to some dimensions than to all others. The first group of strengths were hope, zest, and gratitude, which were the strengths most related to seven, five, and four of the nine PERMA+4 dimensions, respectively. These strengths have previously been identified as "happiness strengths" (along with curiosity and love, [40]) and may indeed represent common correlates of well-being, across different indicators–perhaps similar to the concept of transdiagnostic markers of psychopathology [41]. On the other hand, teamwork appeared to be specifically related with positive relationships (i.e., correlated more strongly than with any other dimension). This was also the case in a previous examination of character strengths and PERMA dimensions [12], and may reinforce the well-known notion that healthy interpersonal relationships are built on trust and cooperation (rather than competitiveness,

[42]). A number of strengths (bravery, honesty, judgment, leadership, perseverance, perspective, prudence) and the general factor of character appeared to be specifically related to accomplishment. These strengths were not as clearly related to this dimension in Wagner et al. [12] and one may speculate that they may rather be strengths that are particularly valued in the organizational context, and therefore especially related to employees' sense of achievement.

The results of the path models shed light on the respective relationships of character strengths (Model 1) and specific strengths and the general factor of character (Model 2) with life satisfaction and positive functioning at work. Model 1 showed that hope, zest, and (to a lesser extent) spirituality were the only three strengths positively related to positive functioning at work. Gratitude, curiosity, and leadership were instead the only three strengths positively related to life satisfaction. These results further strengthen the correlational findings that the "happiness strengths" being the most consistently related to well-being [32, 42], consistent with our expectations (H1). These results suggest that only a few single character strengths may represent direct and indirect pathways to well-being. When all 24 strengths are considered simultaneously through latent factor scores, single character strengths seem to lose their predictive power, with very few exceptions. This may be due to the strong inter-correlations between the 24 strengths (e.g., over .70 for gratitude, hope, and zest), but it also calls into question the role of individual strengths with respect to outcomes such as life satisfaction and positive functioning at work, suggesting that the importance of strengths may lie in what they share. Indeed, most of the variance that strengths share is partialed out multiple regressions are run, as in this case, consequently losing a potentially important part of what character strengths represent or affect.

Our results on the general factor of character support this possibility. The results for Model 2 indicated that the general factor of character was significantly related to both positive functioning at work and life satisfaction. Interestingly, when the general factor of character was included among the predictors, the relationship between positive functioning at work and life satisfaction dropped quite significantly (from .48 to .15). In Model 2, only specific zest, hope, gratitude, perseverance, and self-regulation resulted as positively related to positive functioning at work, and only specific gratitude, hope, zest, love, and self-regulation to life satisfaction. Again, these were primarily happiness strengths [40], together with self-regulation and perseverance, reinforcing the possibility that these two may be specific to the organizational context. Taken together, these findings suggest that having a good character in general (without the need to prescribe specific strengths to account for it) may help build the elements of well-being and ultimately promote life satisfaction. This general factor of character may represent an underlying positive attitude towards life, as previously suggested by other authors [19], which captures what all character strengths have in common. Nurturing character would then mean training all character strengths together, relying on the evidence that it is this general positivity that then explains the positive effects on both the work-related elements of well-being and general life satisfaction.

In line with our hypothesis (H2), positive functioning at work was positively related to life satisfaction in both models, consistent with previous findings suggesting that these nine dimensions represent pathways for building well-being (15). This confirms that employees who express a higher satisfaction with their life are those who also experience more positive emotions, feel more engaged, have better relationships, see meaning in their work, and feel they can achieve their desired goals, while also feeling better physically, having a growth mindset, enjoying their environment, and not worrying about money. Thus, the present results replicate the findings of Donaldson and colleagues and provide further support for their model, while also contributing to the debate sparked by Goodman et al. [13] on the nature and significance of Seligman's [6] PERMA model. More specifically, our results provide initial support

for Seligman's claim that PERMA dimensions are better understood as pathways to reach well-being, or as "psychological ingredients" that constitute well-being, rather than representing a distinct type of well-being, to be separated from subjective or psychological well-being.

Of note, we also observed several small negative associations between specific character strengths and the two outcomes, possibly because the general factor of character (what character strengths have in common) carries the most explanatory power. However, when looking at the mediating effect of positive functioning at work though, we found several small positive indirect effects (H3), suggesting that some specific strengths may still contribute to a more positive work functioning, which in turn may lead to greater life satisfaction.

All in all, why are strengths related to life satisfaction? What is the process that links strengths to such a conceptually broad and far-reaching outcome? Our results show that the PERMA+4 dimensions, that represent positive functioning at work, may bridge this gap and represent work-related pathways to life satisfaction that are more frequently adopted by people who show higher character strengths. Overall, it appears that strengths, especially as general character, may play a prominent role in building work-related PERMA dimensions, and that PERMA+4 dimensions are in turn relevant to overall life satisfaction. These findings have practical implications, at the organizational and individual levels. They suggest that promoting the awareness and development of PERMA+4 elements in organizational contexts may have effects on the employees life satisfaction, and that helping them be aware, explore, and apply [43] their character qualities through formal (character strengths-based interventions) and informal (by highlighting them as core organizational values) practices may further strengthen their well-being.

The present study also has some limitations. First, the cross-sectional nature of the study does not allow causality to be inferred. Future studies should examine these relationships longitudinally, to assess whether character strengths and PERMA+4 actually predict life satisfaction over time, not just cross-sectionally. Second, we only focused on domain-general satisfaction, and only on the cognitive component of subjective well-being. Future work should also consider domain-specific forms of satisfaction, such as job satisfaction, and positive and negative affect as a relevant outcome that may be associated with both character strengths and PERMA+4. In addition, studies are needed to assess whether the PERMA+4 dimensions mediate the efficacy of strengths interventions, thereby further elucidating the mechanisms linking the "building blocks of well-being" to well-being itself. In this regard, there is some evidence that working with strengths and mindfulness positively impact PERMA dimensions in working undergraduates [44].

In summary, the present study newly examined character strengths as the building blocks of the building blocks (i.e., PERMA+4 dimensions) of life satisfaction, and found evidence that character as a whole, and the happiness strengths (gratitude, hope, zest, curiosity and love) may support employees' PERMA+4 dimensions and, indirectly, overall life satisfaction.

## Supporting information

**S1 Table. Overview of the presented well-being theories.**
(DOCX)

**S2 Table. The VIA classification.** Adapted from Peterson and Seligman (2004, pp. 29–30).
(DOCX)

**S3 Table. Linear multiple regression with PERMA+4 and character strengths as predictors, and life satisfaction as outcome.** *Note. ß* = standardized beta coefficient; * *p* < .05, ** *p* < .01,

*** $p < .001$.
(DOCX)

## Author Contributions

**Conceptualization:** Nicole Casali, Tommaso Feraco.

**Data curation:** Nicole Casali.

**Formal analysis:** Nicole Casali.

**Methodology:** Nicole Casali, Tommaso Feraco.

**Supervision:** Tommaso Feraco.

**Writing – original draft:** Nicole Casali.

**Writing – review & editing:** Tommaso Feraco.

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
