## [Decision Letter · Decision Letter 0]

2 Aug 2024

PONE-D-24-19439Which character strengths may build organizational well-being? Insights from an international sample of workersPLOS ONE

Dear Dr. CASALI,

Thank you for submitting your manuscript to PLOS ONE. After careful consideration, we feel that it has merit but does not fully meet PLOS ONE’s publication criteria as it currently stands. Therefore, we invite you to submit a revised version of the manuscript that addresses the points raised during the review process.

Both reviewers considere that the manuscript has merit, but they both also raise some questions, and make some sugestions. We invite authors to address each comment raised by the reviewers, and to revised the manuscript accordingly. 

We look forward to receiving your revised manuscript.

Kind regards,

Paulo Alexandre Soares Moreira, PhD

Academic Editor

PLOS ONE

Journal Requirements:

Reviewers' comments:

Reviewer's Responses to Questions

**Comments to the Author**

1. Is the manuscript technically sound, and do the data support the conclusions?

Reviewer #1: Partly

Reviewer #2: Yes

2. Has the statistical analysis been performed appropriately and rigorously? 

Reviewer #1: Yes

Reviewer #2: Yes

3. Have the authors made all data underlying the findings in their manuscript fully available?

Reviewer #1: Yes

Reviewer #2: Yes

4. Is the manuscript presented in an intelligible fashion and written in standard English?

Reviewer #1: Yes

Reviewer #2: Yes

5. Review Comments to the Author

Reviewer #1: Which character strengths may build organizational well-being? Insights from an international sample of workers

The present study analyses character strength (24 strength and a general factor) in relation to well-being (PERMA+4) and satisfaction in a large sample of 5,487 employees. The manuscript quotes the relevant articles and develops toward the 4 research questions posed. Next to zero order correlations also a mediation analysis was performed to see whether PERMA+4 mediates the relation between character strength and life satisfaction.

Regarding the zero order correlations it is not surprising that strengths correlate more highly with the PERMA total score than life satisfaction does. This could be due to both higher reliability (due to higher number of items) and broader content (which reduces the number of low correlations. So the statement that “we found that 21 of the 24 character strengths were more strongly correlated with PERMA+4 than with life satisfaction” needs to be reevaluated. Likewise, the authors should consider using strength and PERMA in a normal regression analysis to acknowledge and that you have cross-sectional data. You say that you need a longitudinal design but as long as you don’t have one it is safer to avoid mediation analysis. Maybe predictors (and an interaction) would also yield interesting results.

A few minor questions:

You talk about SWB (which included life satisfaction, and affect balance) but you only measure satisfaction with life? Why not including positive and negative affect as well? Or avoid SWB altogether?

You use the term “building block” often, even “building blocks of the building blocks”. Can you elaborate what building block implies or means. As far as I know it is not a technical term in psychology, nor a widely used expression with a clear meaning.

You write “It is historically distinct from psychological well-being (PWB), which instead represents the eudaimonic type of well-being, i.e., a focus on growth and optimal psychological functioning, with a greater attention to the interpersonal dimension (4).” SWB was there before PWB; therefore it is misleading or at least imprecise to say it Is “historically distinct”.

You write “In both models, life satisfaction (modelled as a latent variable) was the dependent variable, and Positive Functioning at Work (also modelled as a unidimensional latent variable) was the mediator.”

“Table 5. Results of the first path model (M1), with character strengths as predictors, Positive

Functioning at Work as mediator, and life satisfaction as dependent variable“ Why did you nt consider work satisfaction?

Reviewer #2: Dear Authors,

The article presents a relevant theme, given that the construct of well-being has been the object of studies and research aimed at understanding the phenomenon and, consequently, subsidizing the construction of public policies, as well as organizational policies and guidelines, in order to favours people's happiness in various dimensions of their lives. Research on the PERMA model is still scarce, which gives the study an innovative character.

The article presents a number of positive points, namely:

-The title fully corresponds to the study presented, in addition to presenting an adequate number of words.

-The abstract presents the required elements, such as objectives, methodology, scope, sample, main results and conclusions.

-The introduction denotes conceptual elements necessary for the formulation of the problem, as well as evidencing the objectives of the study. Elements that show the relevance of the subject addressed are also presented. This paper presents a review of the current literature.

- The large international sample.

-The methodology is consistent with the study's proposal. Statistical analyses are appropriate to the proposed objectives.

However, it is recommended that authors make some changes to the article for publication.

The suggestions for change/improvement are as follows:

- In the introduction, Table 1(Overview of the presented well-being theories). and Table 2 (The VIA classification. Adapted from Peterson 156 and Seligman ((2), pp. 29-30) should be removed.

- In the introduction, between lines 135 to 154, authors must present the information in a narrative form and not by topics.

- In the introduction, it would be relevant for the authors to present the concept/perspective of organizational well-being inherent to their research;

- The ethical procedures of the investigation must be better explained.

- In the results there is some repetition between the data presented in Table 4. (Correlations of character strengths, virtues, and character with PERMA+4) and the text between Line 281 and Line 301. Thus, it is suggested to synthesize the text between Line 281 to Line 301 and only the most relevant correlations are mentioned in the text.

- Authors should present the limitations inherent to their research.

- In the discussion, it is important that the authors present the implications of their research for the development of policies and practices that promote organizational well-being.

In general, I would like to highlight the interest of the work submitted for publication and I congratulate the authors on their work. However, I believe that these changes are necessary to improve its final quality and make it attractive to readers.

6. PLOS authors have the option to publish the peer review history of their article (what does this mean?). If published, this will include your full peer review and any attached files.

Reviewer #1: No

Reviewer #2: No

---

## [Author Response · Author response to Decision Letter 0]

17 Sep 2024

Please see the Response Letter attached

---

## [Decision Letter · Decision Letter 1]

16 Oct 2024

Which character strengths may build organizational well-being? Insights from an international sample of workers

PONE-D-24-19439R1

Dear Dr. CASALI,

We’re pleased to inform you that your manuscript has been judged scientifically suitable for publication and will be formally accepted for publication once it meets all outstanding technical requirements.

Kind regards,

Paulo Alexandre Soares Moreira, PhD

Academic Editor

PLOS ONE

Additional Editor Comments (optional):

Reviewers' comments:

Reviewer's Responses to Questions

**Comments to the Author**

1. If the authors have adequately addressed your comments raised in a previous round of review and you feel that this manuscript is now acceptable for publication, you may indicate that here to bypass the “Comments to the Author” section, enter your conflict of interest statement in the “Confidential to Editor” section, and submit your "Accept" recommendation.

Reviewer #1: All comments have been addressed

Reviewer #2: All comments have been addressed

2. Is the manuscript technically sound, and do the data support the conclusions?

Reviewer #1: Yes

Reviewer #2: Yes

3. Has the statistical analysis been performed appropriately and rigorously? 

Reviewer #1: Yes

Reviewer #2: Yes

4. Have the authors made all data underlying the findings in their manuscript fully available?

Reviewer #1: Yes

Reviewer #2: Yes

5. Is the manuscript presented in an intelligible fashion and written in standard English?

Reviewer #1: Yes

Reviewer #2: Yes

6. Review Comments to the Author

Reviewer #1: thank you for addressing all my concerns in a satisfying way. i have no further questions. i will be of interest to readers.

Reviewer #2: (No Response)

7. PLOS authors have the option to publish the peer review history of their article (what does this mean?). If published, this will include your full peer review and any attached files.

Reviewer #1: No

Reviewer #2: No

---

## [Editor Report · Acceptance letter]

21 Oct 2024

PONE-D-24-19439R1 

PLOS ONE

Dear Dr. CASALI, 

I'm pleased to inform you that your manuscript has been deemed suitable for publication in PLOS ONE. Congratulations! Your manuscript is now being handed over to our production team.

Kind regards, 

on behalf of

Professor Paulo Alexandre Soares Moreira 

Academic Editor

PLOS ONE